# Prognostic Role of Sonographic Decongestion in Patients with Acute Heart Failure with Reduced and Preserved Ejection Fraction: A Multicentre Study

**DOI:** 10.3390/jcm12030773

**Published:** 2023-01-18

**Authors:** Nicola R. Pugliese, Matteo Mazzola, Giulia Bandini, Greta Barbieri, Stefano Spinelli, Nicolò De Biase, Stefano Masi, Alberto Moggi-Pignone, Lorenzo Ghiadoni, Stefano Taddei, Rosa Sicari, Peter S. Pang, Marco De Carlo, Luna Gargani

**Affiliations:** 1Department of Clinical and Experimental Medicine, University of Pisa, 56126 Pisa, Italy; 2Department of Surgical, Medical and Molecular Pathology and Critical Care Medicine, University of Pisa, 56126 Pisa, Italy; 3Department of Experimental and Clinical Medicine, Azienda Ospedaliera Universitaria Careggi, 50134 Florence, Italy; 4Institute of Clinical Physiology—National Research Council, 56124 Pisa, Italy; 5Department of Emergency Medicine, Indiana University, Indianapolis, IN 46617, USA; 6Cardiothoracic and Vascular Department, Azienda Ospedaliero-Universitaria Pisana, 56124 Pisa, Italy

**Keywords:** lung ultrasound, B-lines, pulmonary congestion, acute heart failure

## Abstract

Background: We investigated the role of the dynamic changes of pulmonary congestion, as assessed by sonographic B-lines, as a tool to stratify prognosis in patients admitted for acute heart failure with reduced and preserved ejection fraction (HFrEF, HFpEF). Methods: In this multicenter, prospective study, lung ultrasound was performed at admission and before discharge by trained investigators, blinded to clinical findings. Results: We enrolled 208 consecutive patients (mean age 76 [95% confidence interval, 70–84] years), 125 with HFrEF, 83 with HFpEF (mean ejection fraction 32% and 57%, respectively). The primary composite endpoint of cardiovascular death or HF re-hospitalization occurred in 18% of patients within 6 months. In the overall population, independent predictors of the occurrence of the primary endpoint were the number of B-lines at discharge, NT-proBNP levels, moderate-to-severe mitral regurgitation, and inferior vena cava diameter on admission. B-lines at discharge were the only independent predictor in both HFrEF and HFpEF subgroups. A cut-off of B-lines > 15 at discharge displayed the highest accuracy in predicting the primary endpoint (AUC = 0.80, *p* < 0.0001). Halving B-lines during hospitalization further improved event classification (continuous net reclassification improvement = 22.8%, *p* = 0.04). Conclusions: The presence of residual subclinical sonographic pulmonary congestion at discharge predicts 6-month clinical outcomes across the whole spectrum of acute HF patients, independent of conventional biohumoral and echocardiographic parameters. Achieving effective pulmonary decongestion during hospitalization is associated with better outcomes.

## 1. Introduction

Pulmonary congestion (PC) is one of the main features of patients with heart failure (HF). The increase in extravascular lung water (EVLW) is usually the consequence of the hemodynamic congestion and precedes the overt phase of clinical congestion, when high left ventricular (LV) filling pressures are associated with signs and symptoms of HF [1]. Although it is difficult to establish how much congestion is related to fluid redistribution or volume overload [2], many studies suggest that worsening/occurrence of congestion represents the primary pathophysiological mechanism of acute HF (AHF) [1,3]. Indeed, the evidence of clinical congestion in patients hospitalized for AHF is related to a poor prognosis, expressed by an increased risk of both in-hospital adverse events and short- and medium-term mortality [1]. Guidelines recommend daily evaluation of the signs and symptoms of congestion, fluid balance, vital signs, body weight and kidney function in hospitalized AHF patients, and adjustment of decongestive therapy accordingly [4]. However, clinical evaluation displays a low sensitivity and poor predictive value, as well as other non-invasive imaging tools (e.g., chest X-ray, nuclear medicine and radiology techniques), which also require ionizing radiation exposure [4]. Lung ultrasound (LUS) provides an indirect but accurate measurement of EVLW through the visualization of B-lines both at admission and discharge [5,6,7]. In particular, the assessment of persistent PC at discharge with B-lines has proved to stratify prognosis in HF patients, predicting adverse outcomes, including readmission for worsening HF [8,9,10]. However, little is known about the prognostic role of the reduction in the number of B-lines (pulmonary “decongestion”) during hospitalization and their relation to diuretic therapy in and out of hospital [11,12,13]. The aim of this multicentric study was to assess the dynamic changes of PC in terms of in-hospital variations of B-lines, and to evaluate their prognostic role in patients admitted for AHF.

## 2. Materials and Methods

Patient population. We conducted a prospective, multicentre, observational study in adults hospitalized for AHF, regardless of left ventricular ejection fraction (LVEF) (Supplemental Appendix A). Patients were recruited from the inpatient units of four hospitals in Pisa (Cardiology [n = 67] and Internal Medicine [n = 89] Departments), Florence (Internal Medicine Department [n = 61]) and Chicago (Cardiology Department [n = 20]). A definite diagnosis of AHF was based on the 2016 European Guidelines for the diagnosis and treatment of AHF and CHF [4]. Patients were grouped into HF with reduced LVEF (HFrEF) if LVEF was <50%, and HF with preserved LVEF (HFpEF), if LVEF was ≥50% [4]. The exclusion criteria were: moderate-to-severe lung disease defined by pulmonary function tests and/or presence of pulmonary fibrosis, pneumonia or pulmonary malignancy at computed tomography scans, to avoid potential bias in LUS findings; dialysis; pregnancy; NT-proBNP below the age-adjusted cutpoint in the presence of LVEF ≥50% (≤900 pg/mL ages 50–75; ≤1800 pg/mL over age 75) [14]. The local Ethical Committees approved the study. All subjects gave informed consent, and the study was performed in accordance with the ethical standards of the 1964 Helsinki declaration and its later amendments, and with local guidelines for good clinical practice.

Echocardiographic assessment. All patients underwent transthoracic echocardiography examination at rest within 24 h from admission. We used commercially available ultrasound machines (IE33 and CX50 Philips Medical Systems, Andover, MA, USA; Famiglia Mylab25, Esaote, Genoa, Italy; Sonosite M-Turbo ultrasound machine, FUJIFILM SonoSite, Inc., Bothell, WA, USA) equipped with 2.5–3.5 MHz phased-array probes and second harmonic technology. Left ventricular (LV) volumes were measured and LVEF obtained by 2- and 4-chamber view using the biplane discs summation method (modified Simpson’s rule). LV mass was calculated by the Devereux formula and then indexed to body surface area. Tricuspid annular plane systolic excursion (TAPSE) was measured with the M-mode cursor oriented to the junction of the tricuspid valve plane with the right ventricle (RV) free wall. RV—right atrial pressure gradient was derived using the simplified Bernoulli equation from the peak tricuspid regurgitation (TR) velocity. Inferior vena cava (IVC) was reported, and a dilated IVC (diameter > 21 mm) that collapsed < 50% with a sniff was considered abnormal. Valvular regurgitation was qualitatively assessed using color-Doppler, and whenever regurgitation was more than mild, it was quantified according to European Association and Cardiovascular Imaging (EACVI) and American Society Recommendations.

Lung ultrasound. LUS examination at admission (LUS-1) was performed at the time of echocardiography within 24 h of hospitalization by trained investigators using a standardized imaging protocol (28-zone scheme) with the same probe used for the echocardiographic study [15,16]. The patient was scanned in the supine position and, whenever needed to better visualize the lateral chest, in the right or left lateral decubitus for scanning the right or left chest, respectively (Figure 1). In each intercostal space, the transducer orientation was parallel to the ribs, and the number of B-lines was quantified as suggested by international recommendations: when B-lines were clearly distinguishable, they were counted one by one. When they were confluent, the percentage of the white screen compared with the black screen below the pleural line was considered, and then divided by 10 [15,16]. B-lines analysis was performed real-time (two respiratory cycles) and the sum from the 28 scanning sites yielded a score denoting the extent of the extravascular fluid of the lung. Zero was defined as a complete absence of B-lines, while >30 B-lines was considered as severe sonographic PC [17]. A second LUS (LUS-2) was performed before hospital discharge. Different B-lines parameters were evaluated: (1) the absolute number of B-lines at admission and discharge; (2) the difference between B-lines at discharge and admission (ΔB-lines); (3) the percent change in B-lines, i.e., the ratio of ΔB-lines to the number of B-lines at admission (ΔB-lines%); (4) the decongestion rate during hospitalization (ΔB-lines/day), i.e., the ratio of ΔB-lines to the number of days of hospitalization. The LUS inter-observer variability was examined by intraclass correlation coefficient (ICC) before the enrolment on 50 previously acquired LUS videos evaluated by an expert reader (L.G.), using a standardized training protocol [18]. The mean ICC on B-lines number assessment was 0.978 (single measurements, *p* < 0.0001) and 0.989 (average measurements, *p* < 0.0001) between the expert reader and reader 1, and 0.962 (single measurements, *p* < 0.0001) and 0.981 (average measurements, *p* < 0.0001) between the expert reader and reader 2, consistent with previous data [18].

Follow-Up. Clinical and demographic data were taken from medical records. Follow-up data were obtained in all enrolled patients at 180 days after discharge (no missing data). We defined a primary composite endpoint of cardiovascular death and rehospitalization for AHF. The cause of death was elucidated from the medical records, the family, or the physician who signed the death certificate. The definition of cardiovascular mortality required documentation of significant arrhythmias or cardiac arrest, or death attributable to congestive HF or myocardial infarction in the absence of any other precipitating factor. In case of death out of hospital for which no autopsy was performed, sudden unexpected death was attributed to a cardiac cause. AHF rehospitalizations were confirmed through review of electronic medical records, contacting primary care physicians or cardiologists, and through patient follow-up phone calls. Patients were censored at the time of the first event. Follow-up events were adjudicated by two independent trained investigators, blinded to LUS data. In case of disagreement, a third blinded expert was involved in the evaluation.

Statistical Analysis. Continuous measures were expressed as the mean value ± standard deviation or median and interquartile range. Categorical variables were presented as percentages and were compared using the Chi-square test or the Fisher exact test. Student’s *t*-test or Mann-Whitney and Wilcoxon tests were used to assess the differential distribution of data between samples. The correlation coefficient R or Spearman’s rho was assessed when necessary. Cox proportional hazard regression analysis was used to identify predictors of outcome first at univariate and then at multivariate analysis. Variables included in the multivariable model were selected “a priori” based upon pathophysiology. We excluded collinearity using variance inflation factor. The accuracy in predicting the composite endpoint was assessed by receiver operating characteristic (ROC) analysis, reporting the area under the curve (AUC) and the cut-off point having the highest Youden index (sensitivity + specificity − 1). Kaplan-Meier curves were constructed, and log-rank tests were used to test for differences between curves using ROC-derived cut-offs. We estimated the added value of LUS-derived parameters to predict the occurrence of the composite endpoint using the continuous net reclassification index (NRI) and integrated discrimination improvement (IDI). Reclassification was deemed appropriate for participants with events at follow-up moving up in risk category and for participants without events moving down in risk category on the addition of the novel parameter. A *p*-value < 0.05 was used to define statistical significance. All the analyses were carried out with SPSS version 25.0 (IBM Corp., Armonk, NY, USA), except for continuous NRI and IDI statistics (Stata/SE 13.0, College Station, TX, USA).

## 3. Results

### 3.1. Baseline Characteristics

We enrolled a total of 208 consecutive patients; mean age was 76 years (95% confidence interval: 70–84 years), and 75 (36%) were females. The main characteristics of the study population, including demographical, clinical and biohumoral data, are reported in Table 1.

LUS1 was performed at a median of 6 h since admission (IQR: 3–16 h); LUS 2 was performed at a median of 0 days from hospital discharge (IQR: 0–3 days). Median hospital length of stay was 7 days (IQR: 5–13 days), and it was significantly correlated to the NYHA class at admission (Spearman’s rho 0.3, *p* = 0.001), the overall in-hospital diuresis (Spearman’s rho 0.5, *p* < 0.0001) and the intravenous furosemide dosage (Spearman’s rho 0.4, *p* = 0.004). NT-proBNP levels were available in all patients at admission, and in 148/208 (71%) before discharge. During hospitalization, NT-proBNP values significantly decreased (median values from 4325 to 2742 mg/mL, *p* = 0.038). In this case, 25 patients (12%) had no radiographic evidence of PC (vascular congestion, interstitial or alveolar oedema) on admission, whereas sonographic signs of PC were detectable. Here, 20 patients (10%) had an estimated glomerular filtration rate at admission <30 mL/min/1.73 m^2^ (using the Modification of Diet in Renal Disease study equation). Ultrasound parameters from integrated cardiopulmonary ultrasound are displayed in Table 2. About half of the patients (110/208, 53%) had heart valve disease of at least moderate severity, and mitral regurgitation (MR) represented the most frequent valvulopathy (71 patients, 34%). LUS-1 and LUS-2 images were interpretable in all patients (100% feasibility). All AHF patients presented with B-lines at admission, and only 17/208 (8.2%) had no B-lines on LUS-2. In the majority of patients (51.9%) B-lines decreased by at least 50% from admission to discharge, with a median rate of −3 ΔB-lines/day. Patient-reported dyspnoea improved significantly from admission (all patients were discharged with NYHA class ≤2). IVC diameter at admission was significantly correlated with B-lines at admission (Spearman’s rho 0.68, *p* < 0.0001) and at discharge (Spearman’s rho 0.47, *p* < 0.0001), and was significantly related to the length of hospital stay (Spearman’s rho 0.41, *p* = 0.01), overall in-hospital diuresis (Spearman’s rho 0.34, *p* = 0.03) and intravenous furosemide dosage (Spearman’s rho 0.28, *p* = 0.04).

The HFpEF and HFrEF groups were similar with regards to most baseline clinical characteristics, except for older age, lower rate of previous MI and lower NT-proBNP values among HFpEF patients (Table 1). Among ultrasound parameters, a statistically significant difference between groups was noted only for higher LV volumes in HFrEF, and higher relative wall thickness, E-wave and E/A ratio in HFpEF patients (Table 2). All LUS static and dynamic parameters were similar between HFrEF and HFpEF groups.

### 3.2. Clinical Outocomes

At 6-month follow-up, a total of 41 events occurred (Table 1), and the primary endpoint occurred in 38 (18%) patients (3 patients died after rehospitalization for HF). ROC analysis identified a cut-off of >15 B-lines at discharge as the value with the highest AUC in predicting adverse events (AUC 0.80, 0.71–0.83, *p* < 0.0001, sensitivity 75%, specificity 77%, positive predictive value 68%, negative predictive value 87%; Figure 2A).

We then grouped the total population according to that threshold, finding no statistical difference in terms of age, gender, symptoms at admission and cardiovascular risk profile (Table 3).

Patients discharged with >15 B-lines had higher NT-proBNP levels both at admission and discharge, along with more signs of PC on chest X-ray at admission. However, they received similar in-hospital treatment and medications at discharge as compared to patients with ≤15 B-lines. Patients discharged with residual PC at LUS also had a higher prevalence of MR, larger right and left atria, a worse right ventricle-pulmonary vascular coupling and more dilated IVC. In-hospital pulmonary decongestion indexes (ΔB-lines% and ΔB-lines/day) were significantly lower (poor decongestion) in patients with >15 B-lines at discharge vs those with ≤15 B-lines. Kaplan-Meier survival analysis revealed that the cumulative incidence of the primary endpoint was 36% in patients with >15 B-lines at discharge and 6% in patients with ≤15 B-lines (*p* < 0.0001; Figure 2B). A value of ΔB-lines% ≤50% was also able to discriminate between patients with and without events at follow-up (AUC 0.67, 0.59–0.75, *p* < 0.0001, sensitivity 54%, specificity 85%, positive predictive value 34%, negative predictive value 97%, Appendix A). The Kaplan-Meier survival curves (Appendix A) described a significantly better 6-month outcome for patients with ΔB-lines% >50% (effective decongestion), compared to patients with ΔB-lines% ≤50% (poor decongestion; *p* = 0.003). We included the demographic, clinical and ultrasound parameters associated with the primary endpoint at univariate analysis (Appendix A) in a multivariate model (Table 4). B-lines at discharge showed an independent prognostic value, together with other parameters acquired at admission: NT-proBNP, MR and IVC expiratory diameter. The number of B-lines at discharge was the only marker able to predict events both in HFrEF and HFpEF (Table 4).

ΔB-lines% was significantly associated with the primary endpoint only at univariate analysis. However, adding ΔB-lines% ≤50% (poor decongestion) in the model based on B-lines >15 at discharge, event classification significantly improved: 36/170 (21%) patients not experiencing the primary endpoint were reclassified correctly, while 7/170 (4%) were reclassified incorrectly. Appropriate reclassification of patients with respect to the primary endpoint was observed in 1/38 (3%) subjects, while inappropriate reclassification in 1/38 (3%), yielding a continuous NRI of 22.8%, *p* = 0.04. Discrimination also improved, as indicated by IDI: 4%, *p* = 0.01. Kaplan-Meier survival analysis for the primary endpoint according to B-lines at discharge and ΔB-lines% (Figure 3) showed that patients with >15 B-lines at discharge had the worst outcome irrespective of ΔB-lines% (Group 3 and Group 4), followed by subjects with ≤15 B-lines at discharge and ΔB-lines% ≤50% (Group 2). The presence of B-lines at discharge ≤15 and ΔB-lines% > 50% identified the best outcome (Group 1). Cumulative incidence of adverse events was 4% in Group 1, 15.4% in Group 2, 28.6% in Group 3, and 31.8% in Group 4.

## 4. Discussion

To the best of our knowledge, this is the first multicenter study comparing the prognostic role of B-lines at admission and discharge and of the dynamic changes of B-lines in both HFrEF and HFpEF, together with a thorough cardiopulmonary ultrasound evaluation and a comprehensive clinical and biohumoral assessment. Our results underline the importance of identifying residual PC before discharge to predict adverse events (cardiovascular death and rehospitalization for AHF), independent of other key variables usually acquired at admission, including NT-proBNP, the presence of at least moderate MR and IVC diameter. Moreover, B-lines are the only cardiopulmonary ultrasound parameter to be of independent prognostic value in both HFrEF and HFpEF. Next, assessing the dynamic evaluation of pulmonary decongestion during hospitalization could further refine PC status and patient risk stratification.

Much evidence supports the use of B-lines as “point-of-care” ultrasound approach in different settings, from emergency departments to outpatients clinics, for the differential diagnosis of dyspnea of unclear origin, to rule in or rule out HF [5,6,16,19,20,21]. Our findings support a role for LUS beyond diagnosis. In fact, notwithstanding guideline-directed medical therapy, almost 50% of patients admitted with AHF are discharged with significant residual PC, which is associated with rehospitalization and cardiac mortality within 6 months, probably due to the lack of an accurate algorithm for decongestive therapy monitoring [22,23]. In recent years building evidence supports the role of B-lines evaluation at discharge of AHF or during office visits [8,9,10,12,20,24,25,26]. Most of AHF patients are discharged when they are asymptomatic but still with some degree of PC. Indeed, clinical congestion represents the ‘tip of the iceberg’ of the haemodynamic derangements that precede symptoms, passing through the increase in PCWP with consequent EVLW accumulation [1,3]. B-lines could be a useful tool for recognizing those patients “flying below the radar”, without clinical congestion but with residual PC and therefore at risk of HF rehospitalization or adverse outcome [8,9,10,24,25,26]. Noteworthy, HFrEF and HFpEF patients had similar LUS signs of PC at admission, suggesting LV filling pressure is a common finding in these different phenotypes. In addition, HFrEF and HFpEF reached a similar degree of decongestion before discharge, implying that unloading therapy (mainly diuretics) are equally effective, regardless of baseline LVEF. These findings are in line with those obtained in a long-term European registry [27] and should encourage the assessment of residual PC in both phenotypes of AHF. In the current study only 8% of patients had no B-lines before discharge, and the presence of >15 B-lines identified subjects with worse biohumoral profile (higher NT-proBNP levels), higher prevalence of MR, increased central venous pressure (dilated IVC without inspiratory collapse) and more advanced signs of right ventriculo-arterial coupling. All the above-mentioned parameters can be related to the congestive status in AHF and have an independent prognostic role [4,28,29]. Our data confirm the additional prognostic value of B-lines at discharge across the whole spectrum of AHF patients, promoting the inclusion of cardiopulmonary ultrasound in the clinical judgement together with bio-humoral evaluation. Randomized studies are needed to evaluate if a strategy of titrating diuretic therapy to multiple targets before hospital discharge may improve clinical outcomes in AHF. Recently, it has been demonstrated that tailored LUS-guided diuretic treatment of PC reduced the number of decompensations needing an urgent visit and improved walking capacity in patients with HF [30,31]. It will be interesting to evaluate if an integrated approach could prevent major adverse cardiac events and avoid the potential pitfalls related to a strategy based on a single parameter, as observed in the GUIDE-IT based on NT-proBNP [32]. Indeed, plasma concentrations of natriuretic peptides not only reflect the severity of congestion in HF patients, but are also influenced by heart rhythm, renal dysfunction and body mass index [4]. Therefore, cardiopulmonary ultrasound assessment could play a complementary role, promoting the synergistic application of biomarkers and imaging [33].

Clinical perspective. The in-hospital management of AHF is currently based mainly on symptom improvement, physical examination findings, urine output, and weight loss. Unfortunately, these are poor markers of congestion and can mislead the right timing of discharge, with potentially detrimental consequences during follow-up [2,4]. The present study analyzed a multicenter AHF population consisting of both HFrEF and HFpEF, with a comprehensive bio-humoral and cardiopulmonary ultrasound evaluation, including LUS both at admission and before discharge. When the length of hospital stay and therapy are not titrated on a thorough review of the congestive status, a residual subclinical PC might go unnoticed and jeopardize outcomes independent of NT-proBNP levels, of the presence of at least moderate MR, and of IVC diameter at admission. On the contrary, in the absence of a significant residual PC (B-lines at discharge ≤15) no cardiovascular deaths were observed at 6-month follow-up. The high NPV of ΔB-lines% >50% which characterizes effective decongestion, allows for the identification of patients with the lowest risk of adverse events (4% in our population). The efficacy of a strategy based on an integrated analysis of simple and widely available bio-humoral and cardiopulmonary ultrasound parameters during hospitalization for AHF needs to be tested in prospective clinical trials.

Limitations. Firstly, NT-proBNP testing pre-discharge was available only in 148/208 (71%) patients, because the second determination was left to the discretion of each centre. Secondly, 32% of the population had atrial fibrillation, which prevented the conventional analysis of diastolic function (28). In addition, we did not report mitral E/e′ ratio because it was available in <50% of patients. Thirdly, we excluded patients with moderate-to-severe lung disease to increase the sensitivity of LUS findings, since B-lines are not specific for EVLW. Therefore, our findings may not apply to patients with concurrent exacerbations of chronic pulmonary disease. Finally, we used a 28-zone lung imaging protocol, which is more time-consuming than the simplified 4-zone or 8-zone scheme [10,24,34]. However, such extensive lung assessment requires only 3–4 min in addition to the time needed for a resting echocardiogram and assures interpretable images in all patients [15], including obese subjects, in whom a simplified protocol can be deceptive [10].

## 5. Conclusions

B-lines measurement can be easily performed during hospitalization for AHF, both at admission and at discharge. The presence of residual subclinical sonographic PC at discharge predicts cardiovascular death and HF rehospitalization across the whole spectrum of AHF patients, independent of conventional bio-humoral and echocardiographic parameters. Among cardiopulmonary ultrasound markers, B-lines at discharge are the only predictor of events in both HFrEF and HFpEF. The dynamic evaluation of pulmonary decongestion during hospitalization can further improve risk stratification.

## Figures and Tables

**Figure 1 jcm-12-00773-f001:**
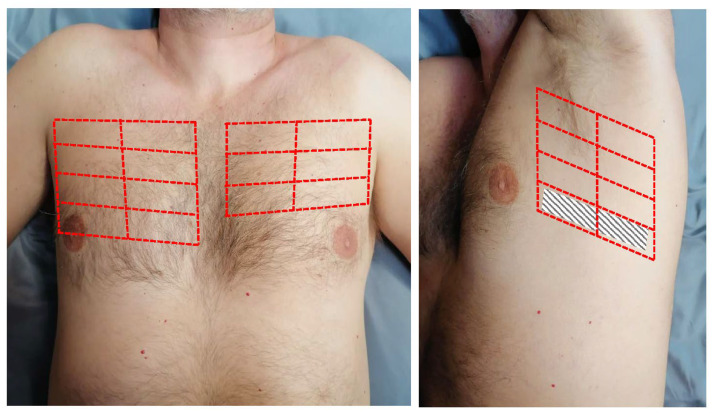
The 28-zone lung ultrasound scanning scheme. The chest is divided along the parasternal, midclavicular, anterior axillary and mid-axillary lines, from the second to the fifth intercostal space on the right hemithorax and from the second to the fourth intercostal space on the left hemithorax, for a total of 16 scanning zones on the right and 12 scanning zones on the left.

**Figure 2 jcm-12-00773-f002:**
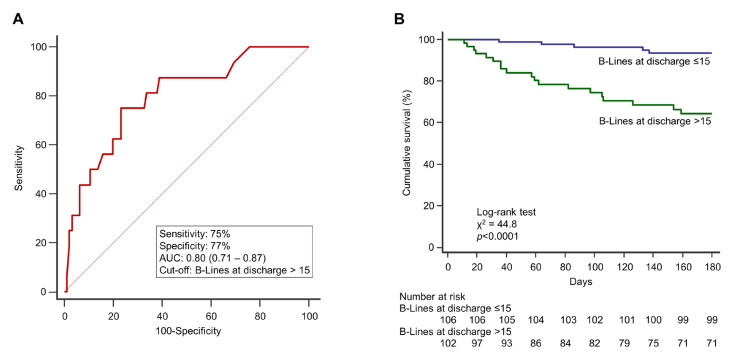
(**A**) ROC curve analysis describing the performance of B-lines at discharge to identify the primary endpoint (rehospitalization for heart failure and cardiovascular death at 6 months). The AUC and 95% confidence interval are shown, as well as the sensitivity, and the specificity at the cut-off identified based on the highest Youden index. (**B**) Kaplan-Meier survival curves stratified according to the ROC-derived cut-off. The curves illustrate a significant difference in cumulative survival, with patients discharged with >15 B-lines experiencing a worse outcome than those having B-lines ≤ 15. Numbers of patients at risk are shown below the survival curves. AUC: area under the curve; ROC: receiver operating characteristic.

**Figure 3 jcm-12-00773-f003:**
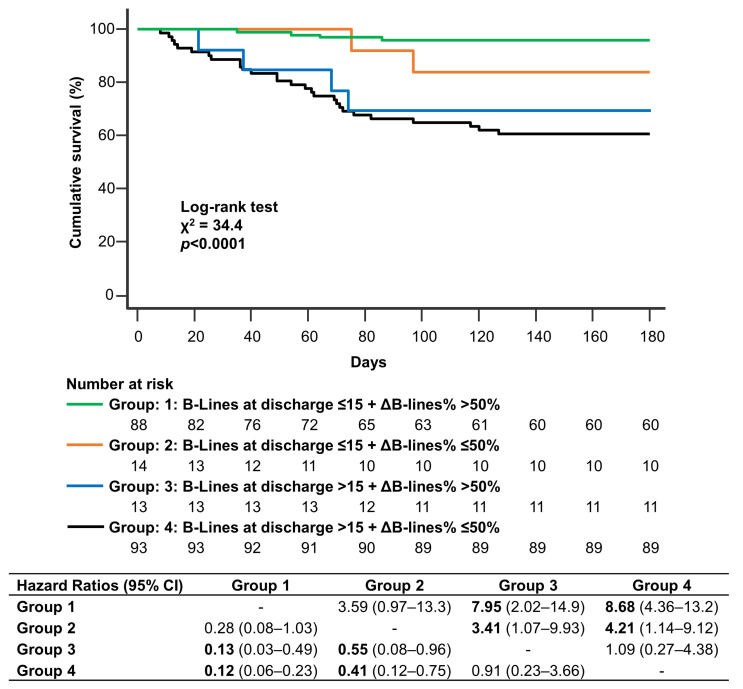
Kaplan-Meier survival analysis for the primary endpoint (cardiovascular death and rehospitalization for heart failure) according to B-lines at discharge and ΔB-lines%. Patients are stratified in four groups according to B-lines at discharge (> 15 or ≤15) and ΔB-lines% (≤50% or >50%). Numbers of patients at risk are shown below the survival curves. Pairwise comparison between groups is shown in accompanying table using hazard ratios and 95% confidence intervals.

**Table 1 jcm-12-00773-t001:** Patient clinical characteristics in the overall population and by left ventricle ejection fraction.

Variable	Total Population (n = 208)	HFrEF (n = 125)	HFpEF (n = 83)	*p*-Value
**Demographics**				
Age, years	75.9 (69.6–83.5)	74 (68.2–80)	79.6 (71.9–86.1)	**0.005**
Female gender	75 (36)	39 (31)	36 (43)	0.1
BSA (m^2^)	1.91 (1.87–1.96)	1.91 (1.84–2.01)	1.87 (1.76–2.02)	0.3
BMI (kg/m^2^)	27.1 (24.7–31.2)	26.2 (26.6–29.5)	26.8 (23.2–31.9)	0.8
Family history of CVD	40 (19)	21 (17)	19 (23)	0.1
Diabetes mellitus	74 (36)	41 (33)	33 (40)	0.1
Arterial hypertension	171 (82)	100 (80)	71 (86)	0.3
Dyslipidaemia ^	80 (37)	50 (40)	30 (36)	0.1
CAD	80 (37)	51 (41)	29 (35)	0.1
Previous MI	73 (35)	53 (42)	20 (24)	**0.01**
Previous coronary revascularization	75 (36)	51 (41)	24 (29)	0.1
Atrial fibrillation	66 (32)	36 (29)	30 (36)	0.4
**In-hospital evaluation**				
NYHA class II at admission	79 (38)	40 (32)	39 (42)	0.2
NYHA class III at admission	69 (33)	45 (36)	24 (29)	0.3
NYHA class IV at admission	60 (29)	40 (32)	20 (24)	0.2
Creatinine (mg/dL) at admission	1.27 (0.99–1.55)	1.29 (1.08–1.55)	1.24 (0.88–1-40)	0.1
eGFR (mL/min/1.73 m^2^) at admission	57 (43.8–74.1)	55.9 (43.7–69.1)	62.1 (49.6–81.5)	0.1
NT-proBNP (pg/mL) at admission	4325 (2021–365)	4601 (2099–9108)	3004 (1322–5644)	**0.03**
NT-proBNP (pg/mL) at discharge *	2742 (1140–6167)	3109 (1228–6111)	2254 (922–6949)	0.3
Admission chest X-ray				
Vascular congestion	183 (88)	111 (89)	72 (87)	0.7
Interstitial edema	154 (74)	95 (76)	59 (71)	0.5
Alveolar edema	25 (12)	17 (14)	8 (10)	0.5
Unilateral pleural effusion	48 (23)	31 (25)	17 (20)	0.5
Bilateral pleural effusion	17 (8)	11 (9)	6 (7)	0.8
Overall in-hospital i.v.diuresis (L)	10.5 (7.4–14.6)	11.3 (8.1–14.6)	9.5 (7.3–14.5)	0.1
Overall in-hospital i.v. furosemide (mg)	340 (190–535)	370 (220–625)	220 (165–480)	0.5
Patients receiving i.v. inotropes	17 (8)	17 (14)	0	**0.001**
Hospital length of stay (days)	7 (5–13)	8 (6–13)	6 (5–13)	0.1
**Home medications at discharge**				
Beta-blockers	148 (71)	94 (75)	54 (65)	0.2
ACE inhibitor/ARB	165 (79)	105 (84)	60 (72)	0.1
MRA	139 (67)	94 (75)	45 (54)	**0.003**
Furosemide	200 (96)	123 (98)	77 (93)	0.1
Furosemide dose (mg/day)	50 (25–125)	50 (25–75)	75 (50–125)	0.1
Thiazide/thiazide-like diuretics	25 (12)	13 (10)	12 (15)	0.4
Digoxin	46 (22)	36 (29)	10 (12)	**0.007**
Calcium-channel blockers	35 (17)	11 (9)	24 (29)	**0.0004**
Amiodarone	13 (6)	11 (9)	2 (2)	0.1
Statins	50 (24)	31 (25)	19 (23)	0.4
Oral anticoagulants	63 (30)	35 (28)	28 (34)	0.4
Antiplatelet drugs	33 (16)	22 (18)	11 (13)	0.4
**Outcomes at 180 days**				
Cardiovascular death	5 (2)	3 (2)	2 (2)	0.8
Re-hospitalization for HF	36 (17)	22 (18)	14 (17)	0.7
Composite end-point	38 (18)	23 (18)	15 (18)	0.8

Data are presented as number and %, mean and 95% confidence interval if normally distributed or median and first and third quartile if not normally distributed. ^ total cholesterol ≥ 200 mg/dL or LDL-C ≥ 130 mg/dL or on lipid-lowering therapy. * available in 148/208 patients (71%). Bold emphasizes significant *p*-values. i.v.: intravenous; ACE: angiotensin-converting enzyme; ACS: acute coronary syndrome; ARB: angiotensin receptor blocker; BMI: body mass index; BSA: body surface area; CAD: coronary artery disease; CRP: C-reactive protein; CVD: cardiovascular disease; eGFR: estimated glomerular filtration rate; HFpEF: heart failure with preserved ejection fraction; HFrEF: heart failure with reduced ejection fraction; MI: myocardial infarction; MRA: mineralocorticoid receptor antagonist; NYHA: New York Heart Association; NT-proBNP: N-terminal prohormone of brain natriuretic peptide.

**Table 2 jcm-12-00773-t002:** The ultrasound parameters in the overall population and by left ventricle ejection fraction.

Variable	Total Population (n = 208)	HFrEF (n = 125)	HFpEF (n = 83)	*p*-Value
**Echocardiography at admission**				
EDV (mL/m^2^)	167 (92–210)	175 (155–210)	160 (92–189)	**<0.0001**
ESV (mL/m^2^)	103 (75–139)	117 (90–150)	95 (71–122)	**<0.0001**
LV ejection fraction (%)	38.5 (28–55)	32.2 (30.5–33.9)	56.9 (55.7–58.2)	**<0.0001**
LVMi (g/m^2^)	147 (111–165)	146 (132–161)	135 (113–157)	0.2
Relative wall thickness	0.34 (0.30–0.40)	0.31 (0.30–0.33)	0.47 (0.40–0.53)	**<0.0001**
LAVi (mL/m^2^)	43.4 (34.2–55.9)	42.4 (32.1–52.4)	45.3 (37.1–58.8)	0.1
Mitral regurgitation *	71 (34)	43 (34)	28 (34)	0.9
Mitral stenosis *	12 (6)	6 (5)	6 (7)	0.8
Aortic regurgitation *	10 (5)	8 (6)	2 (2)	0.3
Aortic stenosis *	17 (8)	9 (7)	8 (9)	0.3
E-wave (cm/s)	95 (86 -120)	96 (89–104)	119 (99–139)	**0.01**
A-wave (cm/s) ^#^	56 (44–86)	59 (52–67)	66 (45–82)	0.1
E/A ratio ^#^	1.62 (1.07–2.51)	1.85 (1.38–2.08)	1.94 (1.53–2.49)	**0.01**
Restrictive pattern ^§,#^	54 (26)	30 (24)	24 (29)	0.4
RA minor axis (cm/m^2^)	2.3 (2.1–2.6)	2.4 (2.2–2.6)	2.3 (2.1–2.6)	0.5
RVOT PLAX diameter (mm)	27 (25–32)	28 (25–30)	29 (26–31)	0.3
TAPSE (mm)	17.7 (16.7–18.9)	17 (16–18)	19 (17–21)	0.2
PASP (mmHg)	44.2 (38.8–49.7)	47.4 (37.7–61.2)	44.2 (28.1–54.1)	0.2
TAPSE/PASP (mm/mmHg)	0.45 (0.38–0.65)	0.38 (0.25–0.58)	0.45 (0.31–0.65)	0.1
IVC expiratory diameter (mm)	19.3 (17.8–21.1)	19.6 (18.2–21.3)	19.4 (18.3–21.1)	0.3
Dilated IVC without collapse **	133 (64)	80 (64)	53 (64)	0.9
**Lung ultrasound**				
B-lines at admission	39 (21–63)	40 (21–63)	39 (22–62)	0.8
B-lines at discharge	15 (5–38)	14 (5–36)	16 (5–39)	0.7
ΔB-lines	18 (4–37)	19 (13–23)	17 (9–24)	0.7
ΔB-lines% (%)	51 (7–83)	50 (18–82)	53 (6–85)	0.6
Decongestion rate (B-lines/day)	3 (0–5)	2 (1–5)	3 (0–6)	0.7

Data are presented as number and %, mean and 95% confidence interval if normally distributed or median and first and third quartile if not normally distributed. * at least moderate severity. ^§^ E/A ratio >2. ^#^ not available in patients with atrial fibrillation. ** IVC expiratory diameter >21 mm that collapses <50% with a sniff. Bold emphasizes significant *p*-values. EDV: end-diastolic volume; ESV: end-systolic volume; IVC: inferior vena cava; LAVi: left atrial volume index; LV: left ventricle; LAD: left atrial diameter; LVMi: LV mass index; PAPS: pulmonary artery systolic pressure; PLAX: parasternal long axis; RA: right atrium; RVOT: right ventricle outflow tract; TASPE: tricuspid annular plane systolic excursion; TR: tricuspid regurgitation.

**Table 3 jcm-12-00773-t003:** Distribution of clinical and ultrasound parameters considering B-lines at discharge.

Parameter	B-Lines at Discharge ≤ 15 (n = 106)	B-Lines at Discharge > 15 (n = 102)	*p*-Value
**Demographics**			
Age, years	75.9 (69.6–82.5)	75.9 (68.2–84.3)	0.7
Female gender	37 (35)	38 (37)	0.6
BSA (m^2^)	1.95 (1.88–2.01)	1.88 (1.81–1.94)	0.1
BMI (kg/m^2^)	28.6 (24.4–32.7)	26.6 (25.5–29.4)	0.3
Family history of CVD	18 (17)	22 (21)	0.1
Diabetes mellitus	38 (37)	36 (36)	0.9
Arterial hypertension	90 (85)	81 (79)	0.4
Dyslipidaemia	40 (38)	40 (39)	0.1
CAD	38 (36)	42 (41)	0.7
Prior MI	34 (32)	39 (38)	0.5
Prior coronary revascularization	37 (35)	38 (37)	0.7
Atrial fibrillation	32 (30)	34 (33)	0.7
**In-hospital evaluation**			
NYHA class II at admission	40 (38)	35 (34)	0.4
NYHA class III at admission	35 (33)	34 (33)	0.9
NYHA class IV at admission	31 (29)	34 (33)	0.5
Creatinine (mg/dL)	1.20 (0.90–1.41)	1.30 (1.08–1.69)	0.06
eGFR (mL/min/1.73 m^2^) at admission	57.9 (47.9–80.1)	55.1 (40.9–70.1)	0.1
NT-proBNP (pg/mL) at admission	3434 (1618–7127)	5989 (2997–9470)	**0.005**
NT-proBNP (pg/mL) at discharge *	1680 (1267–2999)	3166 (2585–6724)	**0.0007**
Admission chest X-ray			
Vascular congestion	92 (87)	91 (89)	0.7
Interstitial edema	71 (67)	83 (81)	**0.02**
Alveolar edema	6 (6)	19 (19)	**0.01**
Unilateral pleural effusion	13 (12)	36 (35)	**0.001**
Bilateral pleural effusion	8 (7)	9 (9)	0.9
In-hospital diuresis (L)	10.5 (7.3–14.7)	9.5 (7.5–14.5)	0.7
Intravenous furosemide (mg)	340 (190–555)	335 (160–500)	0.6
Intravenous inotropes	8 (7)	9 (9)	0.7
Hospital length of stay (days)	7 (5–11)	8 (5–15)	0.3
**Home medications**			
Beta-blockers	71 (67)	73 (71)	0.4
ACE inhibitor/ARB	76 (72)	74 (73)	0.8
MRA	68 (64)	71 (71)	0.4
Furosemide	101 (95)	99 (97)	0.9
Furosemide dose (mg/die)	50 (25–75)	75 (50–125)	0.1
Thiazide/thiazide-like diuretics	12 (11)	13 (13)	0.7
Digoxin	22 (21)	24 (24)	0.6
Calcium-channel blockers	21 (20)	14 (14)	0.2
Amiodarone	5 (5)	8 (7)	0.6
Statins	24 (23)	26 (26)	0.8
Oral anticoagulants	30 (28)	33 (32)	0.7
Antiplatelet drugs	16 (15)	17 (17)	0.6
**Echocardiography at admission**			
EDV (mL/m^2^)	163 (97–201)	170 (113–216)	0.2
ESV (mL/m^2^)	97 (71–124)	111 (82–145)	0.1
LV ejection fraction (%)	40 (30–50)	34.5 (25–55)	0.6
LVMi (g/m^2^)	146 (104–157)	148 (113–170)	0.5
Relative wall thickness	0.34 (0.31–0.40)	0.33 (0.30–0.42)	0.8
LAVi (mL/m^2^)	41.4 (32.6–53.9)	44.5 (35.2–57.6)	**0.002**
Mitral regurgitation	28 (27)	43 (42)	**0.03**
Mitral stenosis	4 (4)	8 (8)	0.2
Aortic regurgitation	5 (5)	5 (5)	0.9
Aortic stenosis	10 (9)	7 (7)	0.4
E-wave (cm/s)	100 (89–130)	93 (83–119)	0.4
A-wave (cm/s)	50 (42–88)	66 (49–81)	0.4
E/A ratio	1.61 (1.07–2.16)	1.90 (1.12–2.77)	0.4
Restrictive pattern ^§^	29 (28)	25 (24)	0.8
RA minor axis (cm/m^2^)	2.3 (2.1–2.5)	2.4 (2.1–2.7)	0.3
RVOT PLAX diameter (mm)	28 (25–31)	27 (24–32)	0.9
TAPSE (mm)	18 (16–22)	16 (14–19)	**0.02**
PASP (mmHg)	34.9 (24.5–42.2)	50.4 (40.2–61.6)	**0.04**
TAPSE/PASP (mm/mmHg)	0.57 (0.40–0.69)	0.38 (0.27–0.45)	**0.03**
IVC expiratory diameter (mm)	17.6 (16–19.2)	22.1 (19.1–25.2)	**0.009**
Dilated IVC without collapse	56 (53)	77 (75)	**0.001**
**Lung ultrasound**			
B-lines at admission	28 (15–44)	43 (22–63)	**0.01**
B-lines at discharge	6 (3–8)	28 (19–45)	**<0.0001**
ΔB-lines	−22 (−36–−8)	−7 (−23–−6)	**<0.0001**
ΔB-lines% (%)	−81 (−90–−60)	−26 (−42–−26)	**<0.0001**
Decongestion rate (ΔB-lines/day)	−4 (−7–−2)	−2 (−4–0)	**<0.0001**
**Outcomes at 180 days**			
Cardiovascular death	0	5 (5)	**0.005**
Re-hospitalization for HF	7 (7)	29 (28)	**<0.0001**
Composite end-point	7 (7)	31 (30)	**<0.0001**

* available in 148/208 patients (71%). ^§^ E/A ratio > 2. Abbreviations as in Table 1 and Table 2. Bold emphasizes significant *p*-values.

**Table 4 jcm-12-00773-t004:** Multivariate predictors of the composite endpoint (cardiovascular death and re-hospitalization for worsening heart failure) at 6-month follow-up.

Parameter	Overall Population (n = 208)		HFpEF (n = 83)		HFrEF (n = 125)	
	HR (95% CI)	*p*-Value	HR (95% CI)	*p*-Value	HR (95% CI)	*p*-Value
NT-proBNP (pg/mL) at admission	**1.00 (1.00–1.01)**	**0.017**	1.00 (0.99–1.01)	0.132	1.00 (0.99–1.01)	0.588
Mitral regurgitation at admission	**3.47 (1.02–11.7)**	**0.042**	4.10 (0.50–3.35)	0.188	4.63 (0.73–2.95)	0.105
IVC expiratory diameter (mm) at admission	**1.15 (1.03–1.29)**	**0.012**	1.11 (0.76–1.62)	0.602	**1.15 (1.06–1.66)**	**0.015**
B-lines at discharge	**1.02 (1.01–1.05)**	**0.023**	**1.08 (1.01–1.16)**	**0.024**	**1.07 (1.02–1.12)**	**0.007**
ΔB-lines%	0.99 (0.99–1.01)	0.281	0.99 (0.99–1.01)	0.398	**0.96 (0.94–0.99)**	**0.009**

Abbreviations as in Table 1 and Table 2. Bold emphasizes significant *p*-values.

## Data Availability

The data presented in this study are available on request from the corresponding author.

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
