# Peer review of "Prognostic Role of Sonographic Decongestion in Patients with Acute Heart Failure with Reduced and Preserved Ejection Fraction: A Multicentre Study"

_jcm, 2023, doi:10.3390/jcm12030773_

Round 1

Reviewer 1 Report

In this study, the authors investigated the role of the dynamic changes of pulmonary congestion, which was assessed by sonographic B-lines, as a tool to stratify prognosis in patients admitted for acute heart failure with reduced and preserved ejection fraction, and concluded the presence of residual subclinical sonographic pulmonary congestion at discharge predicts 6-month clinical outcomes. The design of the study is meticulous, and the detailed information for the study is well-described, as well as statistical analysis. The results of the study is convincing and might be promising in clinical practice. 

minor revision for little mistakes (2 period in line 132, future tense was used in line 136).

Author Response

Point 1. minor revision for little mistakes (2 period in line 132, future tense was used in line 136).

Response 1. Thank you very much for your comments and kind words. We corrected the two minor mistakes, as suggested.

Reviewer 2 Report

The work is interesting, as I addresses the utility of an increasingly widespread practice, bedside ultrasound, to improve care for patients with heart failure.

It is well designed and the results are well communicated.

Here are some corrections and considerations

The methodology for lung ultrasound evaluation is well explained, but reference for the 28-zones protocol should be given. Also, it would be interesting to explain what position the patient should be in when the ultrasound is performed. Perhaps a figure would be helpful.

As strength, I think that the previous validation that has been made of the inter-observer variability is very interesting and gives greater validity to the results obtained.

Table 1:

-        Diuresis (L)- Does it refers to overall in-hospital diuresis?- It should be specified

-        Intravenous furosemide - Does it refers to overall in-hospital intravenous furosemide?- It should be specified

-        Furosemide mg/´die – should be typed ´day´

Table 2

-        Decongestion rate B-lines /´die´– should be typed ´day´

Line 47: reference is made to ´current guidelines`. The reference (4) is from 2016. The latest European guidelines have been published more recently and do not mention performing analyzes every day. The reference could be maintained by avoiding the word ´current´

Line 319-320: ´Our findings confirm that most of AHF patients are discharged when they are asymptomatic…´ This is not a finding of the study, it should be omitted.

I recommend to check self-citations. Some of them are review or opinion articles written by the authors about other works. Perhaps in those cases the reference should be the original text that is commented on, or it should be omitted.

Author Response

Point 1. The methodology for lung ultrasound evaluation is well explained, but reference for the 28-zones protocol should be given. 

Response 1. Thank you very much for your comments and kind words. We added references for lung ultrasound methodology.

Point 2. Also, it would be interesting to explain what position the patient should be in when the ultrasound is performed. Perhaps a figure would be helpful.

Response 2. Yes, we agree. In the revised version of the manuscript, we have specified about the position and added a figure with the scanning scheme (new Figure 1).

Point 3. Table 1:

- Diuresis (L)- Does it refers to overall in-hospital diuresis?- It should be specified

- Intravenous furosemide - Does it refers to overall in-hospital intravenous furosemide?- It should be specified

- Furosemide mg/´die – should be typed ´day´

Response 3. We refer to overall in-hospital diuresis and iv furosemide. We have specified it in the revised version of Table. We replaced "die" with "day"

Point 4. Table 2

-  Decongestion rate B-lines /´die´– should be typed ´day´

Response 4. We replaced "die" with "day"

Point 5. Line 47: reference is made to ´current guidelines`. The reference (4) is from 2016. The latest European guidelines have been published more recently and do not mention performing analyzes every day. The reference could be maintained by avoiding the word ´current´

Response 5. Thank you for highlighting this. We corrected the reference as suggested.

Point 6. Line 319-320: ´Our findings confirm that most of AHF patients are discharged when they are asymptomatic…´ This is not a finding of the study, it should be omitted.

 Response 5. We changed the Discussion as indicated.

Point 7. I recommend to check self-citations. Some of them are review or opinion articles written by the authors about other works. Perhaps in those cases the reference should be the original text that is commented on, or it should be omitted.

Response 7. Yes, thank you. In the revised version of the manuscript we have edited the references, avoiding reviews or opinion papers, and including the original papers.